# Bacterial Colonization of Microplastics at the Beaches of an Oceanic Island, Tenerife, Canary Islands

**DOI:** 10.3390/ijerph20053951

**Published:** 2023-02-23

**Authors:** Cintia Hernández-Sánchez, Ángel Antonio Pestana-Ríos, Cristina Villanova-Solano, Cristopher Domínguez-Hernández, Francisco Javier Díaz-Peña, Cristobalina Rodríguez-Álvarez, María Lecuona, Ángeles Arias

**Affiliations:** 1Department of Preventive Medicine and Public Health, Toxicology, Legal and Forensic Medicine and Parasitology, Health Science Faculty, University of La Laguna (ULL), Campus de Ofra s/n, 38071 Santa Cruz de Tenerife, Spain; 2Institute of Tropical Diseases and Public Health of the Canary Islands, University of La Laguna (ULL), Avda, Astrofísico Fco. Sánchez, s/n°, 38206 San Cristóbal de La Laguna, Spain; 3Departmental Unit of Analytical Chemistry, Chemistry Department, Science Faculty, University of La Laguna (ULL), Avda Astrofísico Fco. Sánchez, s/n°, 38206 San Cristóbal de La Laguna, Spain; 4Department of Animal Biology, Soil Science and Geology, Science Faculty, University of La Laguna (ULL), Avda, Astrofísico Fco. Sánchez, s/n°, 38206 San Cristóbal de La Laguna, Spain; 5Microbiology and Infection Control Service of the University Hospital of the Canary Islands (HUC), 38071 Tenerife, Spain

**Keywords:** microorganisms, microplastics, beach, public health, emerging pathogenic microorganisms

## Abstract

(1) Isolated systems, such as oceanic islands, are increasingly experiencing important problems related to microplastic debris on their beaches. The formation of microbial biofilm on the surface of microplastics present in marine environments provides potential facilities for microorganisms to survive under the biofilm. Moreover, microplastics act as a vehicle for the dispersion of pathogenic organisms, constituting a new route of exposure for humans. (2) In this study, the microbial content (FIO and *Vibrio* spp. and *Staphylococcus aureus*) of microplastics (fragments and pellets) collected from seven beaches of the oceanic island of Tenerife, in the Canary Islands (Spain), was determined. (3) Results showed that Escherichia coli was present in 57.1% of the fragments and 28.5% of the pellets studied. In the case of intestinal Enterococci, 85.7% of the fragments and 57.1% of the pellets tested positive for this parameter. Finally, 100% of the fragments and 42.8% of the pellets analyzed from the different beaches contained *Vibrio* spp. (4) This study shows that microplastics act as reservoirs of microorganisms that can increase the presence of bacteria indicating faecal and pathogenic contamination in bathing areas.

## 1. Introduction

Over the last century, excessive use of plastic has increased exponentially, reaching a total of 390 million tons in 2022 [1]. Massive plastic production coupled with its inadequate management has given birth to new forms of plastic pollution [2,3,4,5], leading to the emergence of microplastics (MPs), particles smaller than 5 mm in their longest dimension, found floating in the marine environment at the mercy of the currents [6,7,8,9].

Many islands report increasing problems related to the massive arrival of microplastics at their beaches, as indicated in various studies affected by the North Atlantic gyre concerning Azores [10,11], Madeira [10,12], other Atlantic islands [13,14] and the Canary Islands, [15,16,17,18,19,20,21].

In Europe, the Bathing Waters Directive (BWD), 2006/7/EC, uses FIOs, such as *Escherichia coli* and faecal Enterococci, as key parameters to monitor and control the quality of bathing waters [22]. However, their presence is not analyzed in the MPs deposited on beaches, despite their providing these microorganisms with a more durable substrate than their natural reservoirs (floating marine waste). The formation of microbial biofilms on MPs that can resist between one and two weeks in aquatic environments has been confirmed. Zettler et al. described the ‘plastisphere’, and they highlighted the potential for marine microplastics to host distinct communities of microbes on their surfaces, indicating that plastisphere communities are distinct from surrounding surface water, implying that plastic serves as a novel ecological habitat. Plastic has a longer half-life than most natural floating marine substrates and a hydrophobic surface that promotes microbial colonization and biofilm formation, differing from autochthonous substrates in the upper layers of the ocean [23]. This is not a new topic, Carpenter et al., alerted about this issue in 1972, advised that the polyethylene spheres found in the coastal water of southern New England have bacteria on the surface [24].

The set of microorganisms embedded in the biofilm covering plastic particles can be attached to other microorganisms different from those of the surrounding water, which may facilitate the survival of indicator bacteria and human pathogens, increasing human exposure routes by providing a diffusion vehicle around coastal waters [23,25,26,27,28]. They can also serve as a long-distance means of transport leading to the propagation of pathogenic bacteria to new areas [26,29,30,31].

Studies are being carried out worldwide on the impact of microorganisms present on the surface of MPs in the marine environment [23,25]. However, few studies, in closed seas or bays, have been published on pathogenic microorganisms in MPs that could reach beaches, endangering the population from a public health perspective [29,32,33]. This paper is the first study of an oceanic island in open sea, to identify the pathogenic microorganisms and indicators of faecal contamination present in the MPs that reach the bathing areas of the island of Tenerife (Canary Islands, Spain).

The aim of this study was to determine the bacterial colonization of MPs deposited on the beaches of the island of Tenerife as well as color distribution of the microplastics and chemical composition.

## 2. Materials and Methods

### 2.1. Study Area and Field Work

The study area included seven different beaches of the island of Tenerife, in the Canary Islands, Spain: Almáciga, Las Teresitas, La Viuda, El Socorro, Playa Grande, Punta del Bocinegro and Puertito de Adeje (see Figure 1 and Table 1 for sampling points and sampling locations characteristics). All 7 beaches are frequented by bathers and/or sportspeople all year round.

Sample collection was carried out during low tide, picking the samples of MPs from the lowest tide line, separating fragments and pellets, all between 1 and 5 mm in their longest dimension.

Both types of MPs (fragments and pellets) were collected directly above the sand with fine forceps, previously sterilized by incineration at 450 °C in the laboratory, and subsequently, introduced in sterile cups that were labelled and taken to the laboratory under refrigeration conditions.

### 2.2. Laboratory Analysis of MPs for Microorganism Detection

The MPs separated into fragments and pellets were introduced in 250 mL of buffered peptone water (Oxoid) for 18–24 h at 37 °C in aerobic conditions. After the incubation period, seeding was performed to isolate and identify the different microorganism studied, both faecal contamination indicators and pathogens.

To detect faecal contamination indicators, faecal Coliforms and *Escherichia coli*, Chromocult^®^ agar (BioMerieux, Marcy l’Etoile, France) was used, which is a differential chromogenic culture medium (*E. coli* colonies acquire a color between dark blue and violet, in contrast with the salmon red color of other colonies of Coliform bacteria). Intestinal Enterococci were detected using Slanetz–Bartley agar base medium (Scharlau, Sentmenat, Spain) and confirmed by kanamycin aesculin azide agar medium (Merck, Darmstadt, Germany) [34].

As to *Staphylococcus aureus*, mannitol salt agar medium (Oxoid, Hamphshire, England) was used. All plates were cultured at 37 °C for 24 h. *Vibrio* spp. were isolated in thiosulfate citrate bile salts sucrose (TCBS) agar (Merck, Darmstadt, Germany) and incubated at 37 °C for 24 h [31,32]. Species were identified using MALDI-TOF automated system (VITEK MS v3.0, BioMérieux, Marcy l’Etoile, France). Mass spectrometry type matrix-assisted laser desorption ionization-time of flight (MALDI-TOF MS) was developed [26,29,30,31]. This technology allows for the identification of micro-organisms directly from colonies of bacteria within a few minutes.

### 2.3. MPs Color Classification

Upon completion of the MPs analysis for the detection of microorganisms, MPs were washed and dried to study color distribution. A systematic semiautomatic method to analyze microplastic colors was used with the reference palette of 120 Pantone colors. This method proposed by Martí et al. in 2020 was useful to estimate the relationship between distance to land, size and color of marine plastic debris, giving a qualitative proxy for the aging of marine plastic samples [35].

The MPs obtained from the seven beaches were superimposed on this palette to allow their sorting based on main colors and hues, prior to their counting.

### 2.4. MPs Composition

A Fourier transform infrared (FTIR) spectrometer Cary 630, equipped with a single reflection diamond attenuated total reflectance (ATR) module (Agilent Technologies, Santa Clara, CA, USA), was used to determine the polymeric nature of a certain number of plastic particles, with a ZnSe beamsplitter and a 1.3 mm diameter thermoelectrically cooled deuterium triglycine sulphate (dTGS) detector. Thirty-two scans per spectrum were applied to obtain FTIR spectra (Happ-Genzel apodization function was applied) at a resolution of 8 cm^−1^ in the range 4000 and 650 cm^−1^. Agilent MicroLab PC FTIR software (version 5.7) was used to identify spectra using polymers libraries. The minimum matching for positive identification, according to the indications of the Guidance of Marine Litter in European Seas of the European Commission [36], was set at quality values ≥ 0.70 over 1.00, which corresponds to a match between 70% and 100% of positive. Though the Guidance of Marine Litter in European Seas of the European Commission indicates that formal identification of the polymer composition is not so critical for larger particles (>500 μm) and that a proportion of 5–10% of all samples <100 μm should be routinely checked, we have considered 16.6% as a reference of analyzed MPs composition.

## 3. Results and Discussion

For the purposes of this study, a total of 687 fragments and 139 pellets from the first tide line of seven beaches were collected, since they are the most found types of MPs on the beaches of the island of Tenerife according to previous studies [15,21,37].

As to the study of color distribution (Figure 2) using the systematic semiautomatic method [35], it was observed that the fragments studied (n = 687) presented an increased tendency to acquire lighter hues. Of this total, 46.87% were white colored (31%) and transparent (15.87%) samples, which are the most abundant in Canary Islands coasts according to other previous studies on the islands [15,17,37,38,39]. Of the remaining 53.13%, the predominant colors were sky blue (10.9%), yellow (9.8%), grey (67.0%), blue (6.4%), green (6.0%), pink (6.0%), violet (3.1%), red (2.6%), cyan (1.0%) and turquoise (0.7%). As for hues, 75.3% of the fragments showed light hues (light), 23.3% presented medium hues (medium) and a mere 1.4% showed dark hues (dark).

Regarding the pellets colors distribution (Figure 3) (n = 139), predominant colors were white (44.6%), transparent (28.1%), yellow (13.7%), brown (12.2%), green (0.7%) and sky blue (0.7%). The pellets’ hues were 98.6% light and 1.4% medium. It should be emphasized that the percentages of yellow and brown add up to 25.9%. The pellets found belong to the light hues most probably since their original color comes from white or transparent but has endured a weathering process in the marine environment towards more yellow or brown hues.

The ageing/weathering process, mainly by photodegradation, is the major cause of fragment whitening and pellet yellowing [38,40,41]. In the study of Rodriguez et al., 2019, 90.6% of the pellets were translucent, probably due the nearby dumping of its particles [32]. The results of this study showed that most fragments present light hues, indicating that they probably have been in the marine environment for a long time.

As to MPs composition, a total of 171 samples (16.6%) were analyzed. The most abundant polymers were polyethylene (85.4%) and polypropylene (11.1%). In the study by Frère, et al., 2018, in the Bay of Brest, MPs made of polyethylene, polypropylene and polystyrene were found [42], similarly to Viršek et al., 2019, in the Adriatic Sea, who signal polyethylene (75%) and polypropylene (9.3%) as main components of MPs [43].

They are the most abundant materials, as shown in Figure 4. The Appendix A, includes information on the chemical composition and infrared spectrum of the MPs found on each beach studied.

Table 2 shows the microorganisms identified in the different samples from the studied beaches, following the experimental procedures previously described. As shown in the table, faecal bacteria were found both in fragments and pellets from all beaches sampled, excepting Almáciga beach, where no microbial contamination was found in pellets.

The highest percentage of positive samples in fragments corresponded to the *Vibrio* spp. with 100%, followed by faecal Coliforms and intestinal Enterococci with 85.7%, *Staphylococcus aureus* 71.4% and *E. coli* 57.1%. As to pellets, 57.1% corresponded to faecal Coliforms and *Staphylococcus aureus*, 42.8% to intestinal Enterococci and *Vibrio* spp. and 28.5% to *E. coli*.

The fragments found on the lowest tide line of Playa Grande, Puertito de Adeje and Punta del Bocinegro showed the presence of all the microorganisms studied. However, only the pellets collected from Las Teresitas beach showed the presence of all the microorganisms studied. Overall, bacterial colonization was found to be higher in fragments than in pellets, possibly due to the former’s rougher and more irregular surface (see Figure 5), facilitating the formation of microorganisms’ biofilms. The degradation of fragments in marine environments generates more roughness than in pellets.

*Vibrio* spp. was isolated in 100% of the fragment samples and 42.8% of the pellets. It was the most abundant species described in the study published by Amaral-Zettler, in 2020, on the ecology of the plastisphere [25]. Furthermore, the study performed by Muniz Silva et al. in 2019, in the Guanabara Bay, in Brazil, confirmed the presence of 59 strains of *Vibrio* spp. (mainly *Vibrio cholerae*, *Vibrio vulnificus* and *Vibrio mimicus*) [33]. In the study conducted by Kirstein et al., 2016, on samples of MPs collected from the North and Baltic seas, the presence of *V. parahaemolyticus* was found, indicating that the occurrence of potentially pathogenic bacteria in marine MPs makes it necessary to urgently carry out detailed biogeographic analyses of marine MPs [28].

In all studied cases from the beaches of Tenerife, *V. alginolyticus* was identified. It should be noted that *Vibrio* spp. are ubiquitous in the marine environment, and the high MPs colonization showed in our study suggests that this species is part of the biofilms covering most plastics found on beaches. These bacteria can frequently infect wounds or ears, mostly of children practicing water activities, as well as adults’ at lower extremities [44,45].

As to faecal contamination indicators, our study focused on those included in the European legislation [22], *E. coli*, and intestinal Enterococci and faecal Coliforms. Faecal Coliforms were found in a higher percentage than *E. coli*, and mostly on fragments rather than on pellets. Various studies indicate that biofilms colonizing the so-called “plastisphere” could also act as a reservoir for FIOs, such as *Escherichia coli*, or pathogen bacteria such as *Vibrio* spp., and the knowledge of *E. coli* colonization and the persistence of MPs should be considered an additional risk to coastal waters pollution [32]. In their study carried out in Scotland, these authors found colonization by *E. coli* and Enterococci in the pellets collected from five public sandy bathing beaches. Other studies performed in various countries found MPs colonization by microorganisms, although MPs samples were collected from the sea surface. Accordingly, Virsek et al. [43] indicate that MPs act as transport vectors of pathogen bacteria for fish, such as Aeromonas salmonicida in the samples collected in the water surface in the north of the Adriatic Sea, on the Slovenian coast. Frère et al. [42] studied MPs bacterial communities in the Bay of Brest (Brittany, France) near an area of intense anthropogenic activity, studying the influence of polymer size and type and finding MPs colonized by various *Vibrio* species. In the study by Silva et al. [33], *E. coli* and *Vibrio* spp. were found in Guanabara Bay waters, Brazil. Finally, Pedrotti et al. [46] found different bacterial species, including *Vibrio parahaemolyticus*, in floating microfibers collected in the Liguria Sea, north-east of the Mediterranean Sea. All the studies mentioned were carried out in closed bays or closed seas.

*S. aureus* is not a FIO, however, various authors propose it to be considered as an indicator of bathing water quality, due to potential problems, such as skin and mucous membranes infections [47,48]. In this study, a higher prevalence of colonization by *S. aureus* was found in fragments than in pellets, although all beaches studied showed MPs colonization by this microorganism.

In the studies mentioned, MPs colonization by different microorganisms was found. However, as indicated by Beloe et al., 2022, in their systematic review about plastic debris as a vector of bacterial diseases, most studies reach only the colonization stage, yet there is a need to carry out structured surveys linked to relevant environmental experiments to understand patrons and processes throughout vectorial stages and to allow a more precise assessment of pathogen risks and its impact on polymers [49]. Microplastics carried by the sea from other geographic areas are ending up deposited on beaches as microplastics were picked from beaches with a large influx of bathers.

There are differences in MPs colonization depending on the beach. In the specific case of Almáciga beach, it was observed that it presented the lowest microorganism contamination compared to the other beaches. It only showed faecal Coliforms, intestinal Enterocci and *Vibrio* spp. in fragments (probably as a result of the roughness they present due to aging) and none in pellets. Almáciga beach has the greatest waves of the seven studied beaches, reaching a wave height of up to 3–4 m during certain periods, thus hampering biofilm formation or permanence, especially in pellets. The Appendix A shows the significant wave height (m) of all beaches sampled, starting a year before this study, for further information on the main wave dynamics.

The beaches with the greatest microbial contamination were Puertito de Adeje, Punta del Bocinegro and Teresitas. They all share the same morphological features of enclosed bays or beaches and less waves. They are also located near wastewater discharges (Table in Appendix A), which may encourage MPs colonization by microorganisms.

Although the MPs may come from faraway areas according to their characteristics (ageing/weathering and roughness), the presence of wastewater discharges near the beaches could be causing the increased presence of FIOs (intestinal Enterococci and faecal Coliforms) which, in synergy with the biofilms covering the MPs, allows the persistence of these bacteria in the environment. Appendix A shows relevant data on the proximity of wastewater discharges to the beaches, including sampling points and nearby dumping points. Several studies have shown that MPs can act as microorganisms’ transporters and they have the capacity to shelter in the biofilm pathogen microorganism and FIOs [29,49]. The results of this study indicate that MPs with more bacterial colonization have been found in closed beaches and beaches with nearby wastewater discharges. It is important to highlight that coastal water of the nearby studied beaches are according to the Bathing Water Directive [22] as the Nayade platform confirms [50]. This study suggests that the biofilm of this MP most likely helped them to create a more stable substrate for different microorganisms, being ready to be transported to other geographical areas. 

This study has some limitations, as it focuses on the bacterial colonization stage of MPs, not considering MPs capacity as transmission vectors of the disease to humans and wildlife. However, these findings suggest that MPs can act as substrate for bacterial biofilm formation (including pathogens), and the presence of colonization is a predisposing factor to infection acquisition.

From a public health point of view, the most significant finding is the presence of faecal bacteria (probably due to waste discharge) and *Vibrio alginolyticus*. This microorganism is considered an emerging pathogen mainly causing soft tissue infections and septicemia, normally associated to marine activities [44,51]. It also poses a threat to aquaculture for being pathogenic for fish, shellfish, or mollusks [26,52,53,54,55,56].

## 4. Conclusions

The most abundant polymers found in this study were polyethylene and polypropylene with an increasing tendency to acquire lighter hues caused by the weathering process. The study on MPs from the seven beaches of Tenerife showed high microbial contamination, especially by *Vibrio alginolyticus* and faecal bacteria. The highest colonization was found in fragments and in the most enclosed beaches with less waves, located near wastewater discharges. As a precautionary measure, the presence of microplastics in beaches and their bacterial colonization should be monitored for being potential indicators of bathing water quality.

Further study is needed to assess the actual health risk to the population posed by the presence of microorganisms in the microplastics reaching beaches, studying the movement, introduction, establishment, and propagation of emerging pathogenic microorganisms adhered to these materials in a new geographical area.

## Figures and Tables

**Figure 1 ijerph-20-03951-f001:**
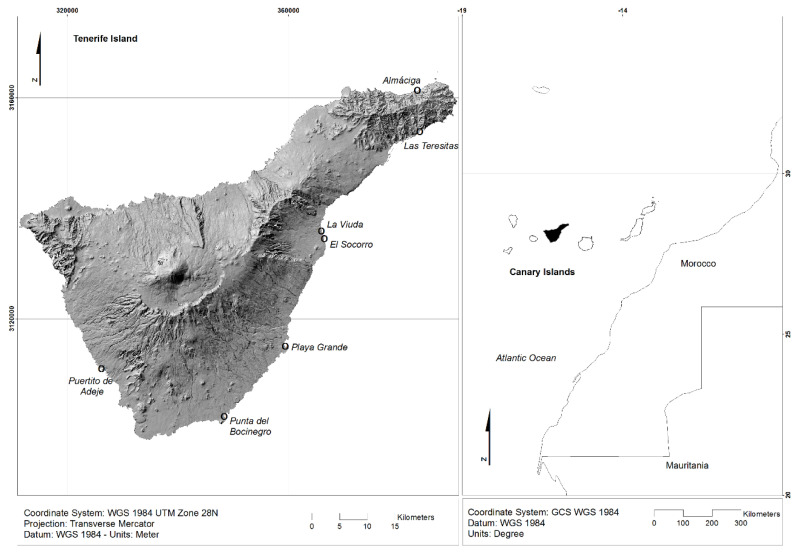
Location of the Canary Islands highlighting the island of Tenerife, located in the Atlantic Ocean and geolocation of the beaches sampled on the island of Tenerife.

**Figure 2 ijerph-20-03951-f002:**
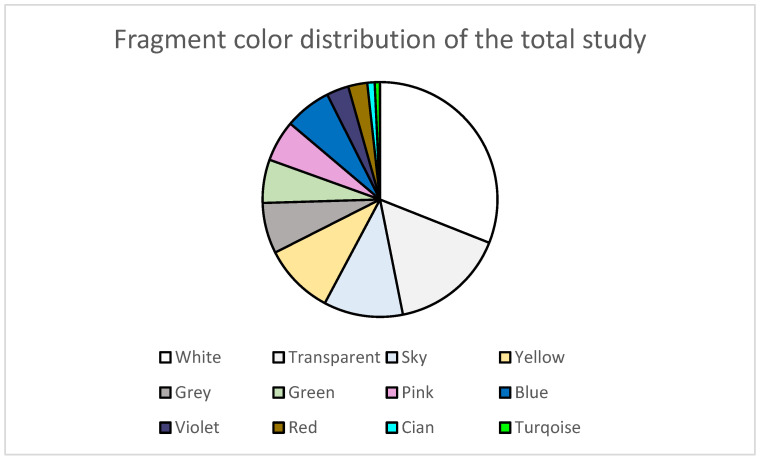
Fragments color distribution of the total study (n = 687).

**Figure 3 ijerph-20-03951-f003:**
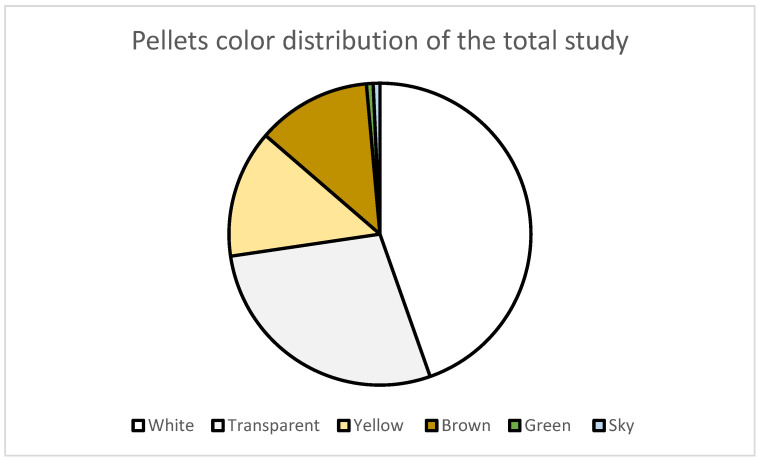
Pellets color distribution of the total study (n = 139).

**Figure 4 ijerph-20-03951-f004:**
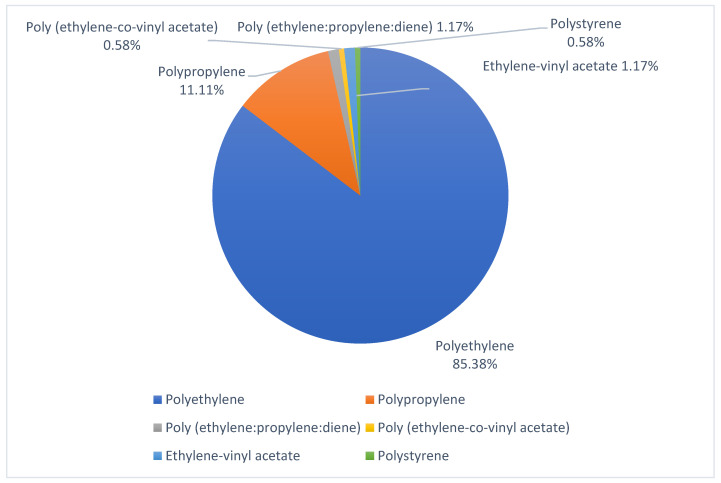
Chemical composition of the total MP analyzed.

**Figure 5 ijerph-20-03951-f005:**
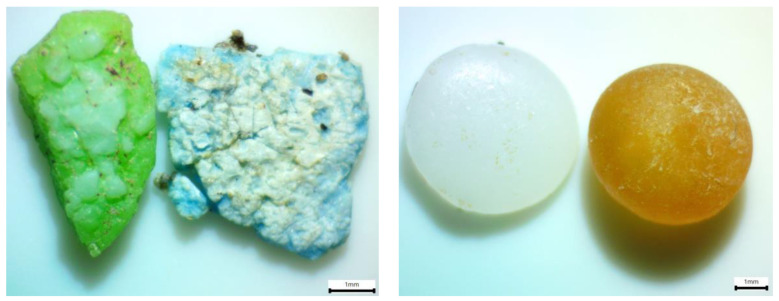
Loupe view of irregularities and roughness presented by MPs found on the beaches of Tenerife. MPs picked from Playa Grande on 21 April 2021. Left image shows high irregularities and roughness of the fragments. Right image shows low irregularities and roughness of the pellets (weathering and not weathering).

**Table 1 ijerph-20-03951-t001:** Data on beaches locations and characteristics, as well as sampling dates.

Beach	UTM Coordinates	Sampling Date	Extension	Orientation	Characteristics
Almáciga	X = 383.337,38 Y = 3.161.160,50	20 April 2021	307 m	N	Black sand and pebbles
Las Teresitas	X = 383.720,42 Y = 3.153.954,60	12 May 2021	1.229 m	SE	White sand
La Viuda	X = 365.963,43 Y = 3.135.761,21	9 May 2021	126 m	E	Black sand and pebbles
El Socorro	X = 366.359,82 Y = 3.134.403,47	11 May 2021	58 m	SE	Black sand
Playa Grande	X = 359.414,00 Y = 3.114.927,65	21 April 2021	180 m	NE	Black sand
Punta del Bocinegro	X = 348.334,87 Y = 3.102.181,66	27 April 2021	286 m	E	Black sand and pebbles
Puertito Adeje	X = 326.313,94 Y = 3.111.047,50	9 May 2021	92 m	SW	Black sand

**Table 2 ijerph-20-03951-t002:** Bacterial content of MPs collected from the seven beaches of the island of Tenerife.

Beach	MP	*E. coli*	Faecal Coliforms	Intestinal Eterococci	*S. aureus*	*Vibrio* spp.
Almáciga	Fragments	-	+	+	-	+
	Pellets	-	-	-	+	-
El Socorro	Fragments	-	-	+	+	+
	Pellets	-	+	+	+	-
Las Teresitas	Fragments	+	+	+	-	+
	Pellets	+	+	+	+	+
Playa Grande	Fragments	+	+	+	+	+
	Pellets	-	-	-	-	-
Puertito de Adeje	Fragments	+	+	+	+	+
	Pellets	-	+	+	-	+
La Viuda	Fragments	-	+	-	+	+
	Pellets	-	-	-	+	-
Punta del Bocinegro	Fragments	+	+	+	+	+
	Pellets	+	+	-	+	+
Percentage of positive	Fragments *	57.1%	85.7%	85.7%	71.4%	100%
	Pellets **	28.5%	57.1%	42.8%	57.1%	42.8%

* Percentage of beaches whose fragments tested positive for the microorganisms studied; ** Percentage of beaches whose pellets tested positive for the microorganisms studied.

## Data Availability

The data presented in this study are available upon request from the corresponding author.

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
