# Peer review of "Bacterial Colonization of Microplastics at the Beaches of an Oceanic Island, Tenerife, Canary Islands"

_ijerph, 2023, doi:10.3390/ijerph20053951_

Round 1

Reviewer 1 Report

I reviewed the present work entitled 'Bacterial colonization of microplastics at beaches of an oceanic 2 island, Tenerife, Canary Islands'.

Now a days the estimation of microplastics has become a great challenge for the researchers. Still the quantification methods need efficient technology. The paper is good. But, I have few questions:

1. How can you say that FTIR has identified all the microplastics only. Organic matter will be there in the sample. So, why you didn't go for the digestion of organic matter. 

2. How bacteria will grow and survive on the Microplastics? In the introduction section, author must add some literature about the bacterial growth and survival on the microplastics. Detail mechanism should be added. This part is missing.

4. Please elaborate methodology section. Justify, why you have used FTIR-ATR for the identification purpose. 

5. Kindly add some clear picture of microplastics which you found in your study.

6. Revise the discussion section with more scientific justification. This part is very important. 

Author Response

I reviewed the present work entitled 'Bacterial colonization of microplastics at beaches of an oceanic island, Tenerife, Canary Islands'.

Now a days the estimation of microplastics has become a great challenge for the researchers. Still the quantification methods need efficient technology. The paper is good. But, I have few questions:

First of all, I would like to thank Reviewer 1 for the time spent reviewing the manuscript as well as for his/her positive comments on it. Regarding his/her specific comments, with we believe have helped to improve our manuscript we have addressed the following.

  1. How can you say that FTIR has identified all the microplastics only. Organic matter will be there in the sample. So, why you didn't go for the digestion of organic matter.

At macroscopy level, there was no visually organic matter because the Atlantic Oceanic at this latitude is oligotrophic. However weathering process make that the plastic particles looks ageing, and the match in FTIR is not correct, for this reason we use cyclohexane in our research to avoid the upper ageing layer and increase the matching.

  1. How bacteria will grow and survive on the Microplastics? In the introduction section, author must add some literature about the bacterial growth and survival on the microplastics. Detail mechanism should be added. This part is missing.

We have introduced in the introduction section some literature about the bacterial growth and survival on the microplastics and the detail mechanism.

  1. Please elaborate methodology section. Justify, why you have used FTIR-ATR for the composition identification purpose.

FTIR-ATR spectroscopy, RAMAN spectroscopy and Pyr GC/MS, are the three different techniques used to identify the composition of the plastic polymers in marine debris. In our previous papers, we have used FTIR-ATR spectroscopy with good matching (>70%), so we have adopted this methodology.

  1. Kindly add some clear picture of microplastics which you found in your study.

The following photos shows 2 pictures of fragments and 2 pictures of pellets.  

  1. Revise the discussion section with more scientific justification. This part is very important.

We have revised the discussion section and added information to clarify the justification.

Reviewer 2 Report

Minor issues

Microplastic contamination is drawing worldwide attention and its effects on the ecosystem remain to be studied. Overall, the manuscript presented data for selected potential pathogenic bacteria distribution, indicating the carrier function of the MPs.

The data in this MS is of environmental significance, but inadequate data collection and description of the MS.

#line70, the season for collection is important for bacteria distribution, so when did the samples were collected, and how about the sea water and local climatic conditions?

#line83, table should be improved for optimized layout, sample date can be in one line

#line143, color distribution should be presented with a pie chart.

#line 107, MS methods are well established and recognized for microorganism identification and classification, but citation of reference and more detailed information is required for the methods section, which might be valuable for readers.

Also, it should be noted that the carrier function of MPs, can be more hazardous if they are capable of transporting pathogenic bacteria over a long distance, or across different regions. This brings out one question of this study, how to differentiate isolated bacteria from local seawater contamination? The bacteria in the seawater can be a blank control for MPs, or the abundance of the bacteria in seawater is low, while compared with a biofilm of MPs which would be much higher.

Author Response

Reviewer 2

Microplastic contamination is drawing worldwide attention and its effects on the ecosystem remain to be studied. Overall, the manuscript presented data for selected potential pathogenic bacteria distribution, indicating the carrier function of the MPs.

 The data in this MS is of environmental significance, but inadequate data collection and Description of the MS.

First of all, I would like to thank Reviewer 2 for the time spent reviewing the manuscript as well as for his/her positive comments on it. Regarding his/her specific comments, with we believe have helped to improve our manuscript we have addressed the following

#line70, the season for collection is important for bacteria distribution, so when did the samples were collected, and how about the sea water and local climatic conditions?

Canary Islands are under the influence of the cold Canary Current (a descending branch of the Gulf Stream) that flows in a southwesterly direction and brings cold waters from more northerly latitudes (and also microplastics). For this reason, the temperature of the sea is lower than it would be due to its latitude. The surface temperature of the waters has low oscillation and vary between 16 – 18ºC minimum in the winter months and 23 – 25ºC in summer. Information about dates sample collection is described in table 1.

#line83, table should be improved for optimized layout, sample date can be in one line

Table has been improved according with the Reviewer 2 suggestion.

#line143, color distribution should be presented with a pie chart.

Graph with color distribution has been presented with a pie chart.

#line 107, MS methods are well established and recognized for microorganism identification and classification, but citation of reference and more detailed information is required for the methods section, which might be valuable for readers

It has been included the citation of references with more detailed information. 

Also, it should be noted that the carrier function of MPs, can be more hazardous if they are capable of transporting pathogenic bacteria over a long distance, or across different regions. This brings out one question of this study, how to differentiate isolated bacteria from local seawater contamination? The bacteria in the seawater can be a blank control for MPs, or the abundance of the bacteria in seawater is low, while compared with a biofilm of MPs which would be much higher.

We have suggested a possible explanation base on the results obtained in this paper and the bibliography consulted and cited in this paper

Round 2

Reviewer 1 Report

It can be accepted now.